# Outcome of acute bacterial meningitis among children in Kandahar, Afghanistan: A prospective observational cohort study

Bilal Ahmad Rahimi[1,2]*, Niamatullah Ishaq[3], Ghulam Mohayuddin Mudaser[4], Walter R. Taylor[5]

1 Department of Pediatrics, Faculty of Medicine, Kandahar University, Kandahar, Afghanistan, 2 Research Unit of Faculty of Medicine, Kandahar University, Kandahar, Afghanistan, 3 Department of Radiology, Faculty of Medicine, Kandahar University, Kandahar, Afghanistan, 4 Department of Histopathology, Faculty of Medicine, Kandahar University, Kandahar, Afghanistan, 5 Mahidol Oxford Tropical Medicine Clinical Research unit (MORU), Mahidol University, Bangkok, Thailand

* drbilal77@yahoo.com

## Abstract

### Background

Acute bacterial meningitis (ABM) is an important cause of morbidity and mortality in children but there are no published data on the treatment outcomes of ABM in Afghanistan.

### Methods

We conducted a prospective observational cohort study over one year, February 2020 to January 2021 in a tertiary care hospital in Kandahar, Afghanistan. AMB was diagnosed clinically and on lumbar puncture findings. Binary logistic regression assessed factors for death.

### Results

A total of 393 ABM children of mean age 4.8 years were recruited. Most were males [231 (58.8%)], living in rural areas [267 (67.9%)] and in households of >10 inhabitants [294 (74.8%)]. Only 96 (24.4%) had received against both *Haemophilus influenzae* type b (Hib) or pneumococcal (PCV) vaccines. Children were treated with combination of ceftriaxone and ampicillin and 169/321 (52.6%) received dexamethasone. Of the 321 children with a known outcome, 69 (21.5%) died. Death was significantly associated with: not receiving dexamethasone [adjusted odds ratio (AOR) 4.9 (95% CI 2.6–9.5, $p$ <0.001)], coma on admission [AOR 4.6 (I 2.3–9.5, $p$ <0.001)], no PCV [AOR 2.8 (1.2–6.6, $p$ = 0.019)] or Hib vaccine [AOR 2.8 (1.2–6.6, $p$ = 0.019)], and being male [AOR 2.7 (1.4–5.5, $p$ = 0.005).

### Conclusions

ABM causes significant morbidity and mortality in Afghan children that may be improved by greater use of PCV and Hib vaccines. Adjunct dexamethasone should be evaluated formally in our setting.

**Data Availability Statement:** All relevant data are within the paper and its Supporting information files.

**Funding:** The authors received no specific funding for this work.

**Competing interests:** The authors have declared that no competing interests exist.

## Introduction

Acute bacterial meningitis (ABM) is a severe disease in children and is associated with mortality in 14.4% (5.3–26.2%) [1] and neurological sequelae in 16–50%; the main sequalae are hearing loss (11%), paresis (4%), seizures (4%), and mental retardation (4%) [2–5]. The pathophysiology of ABM involves inflammation of the blood brain barrier (BBB), loss of vascular autoregulation, increased intracranial pressure and cerebral oedema [6]; hydrocephalus may also occur due to impaired reabsorption of cerebrospinal fluid (CSF) by arachnoid villi and obstructed flow in the third or fourth ventricles [7–9]. All of these factors contribute to neuronal damage and neurological deficits [6].

Globally, in 1990, of the common causes of bacterial meningitis (*Neisseria meningitidis*, *Haemophilus influenzae*, *Streptococcus pneumoniae*) *N. meningitidis* was the leading cause of meningitis related mortality [10]. However, by 2016, previously less common pathogens like *Listeria monocytogenes*, Staphylococcus, Gram-negative bacteria, fungal and viral infections were the leading causes of deaths and incident cases, estimated at 318,400 (95% uncertainty interval 265,218–408,705) and 2.82 million (95% uncertainty interval 2.46–3.31), respectively [10]. This is most probably due to the introduction of PCV and Hib vaccines. The top ten countries for meningitis mortality were India, Nigeria, Ethiopia, Pakistan, Democratic Republic of the Congo, Uganda, Tanzania, Niger, Afghanistan, and China with the four non-African countries, India, Pakistan, Afghanistan, and China reporting 63,001; 16,335; 7,302 and 907 deaths, respectively [10].

Afghanistan has been suffering civil conflict for the last four decades and, together with natural disasters, has weakened its economic development [11], contributing to poor health infrastructure and poor health indices. Afghanistan's public health challenges include poor control of infectious diseases, and limited safe water supply and waste disposal [12]. There is widespread childhood malnutrition and approximately 50% of children have anaemia and vitamin A deficiency [13]. Immunization coverage against routine childhood illnesses is very low with substantial differences between cities and rural areas [14]. The Afghanistan Expanded Program on Immunization (EPI) introduced *Haemophilus influenzae* type b (Hib) in 2009 and the 13-valent pneumococcal conjugate vaccine (PCV13) in 2013 [11]. However, a weak surveillance system precludes an estimation of illnesses or deaths prevented from these two bacterial infections. The schedules for Hib and PCV are 6, 10, and 14 weeks and two doses of measles vaccine are given at 9 and 18 months.

There are no currently no published data on treatment outcomes of paediatric ABM in Afghanistan. We, therefore, set out to assess prospectively the outcome of ABM in children and determine factors associated with death.

## Materials and methods

### Study site

This study was conducted in the paediatric ward of Mirwais Regional Hospital (MRH), a tertiary care regional hospital located in Kandahar city, southwest Afghanistan. Diagnostic capacity is limited as there are no facilities for Gram staining, bacterial and cerebrospinal fluid (CSF) culture. Basic radiology is available.

### Ethical considerations

Written informed consent was obtained from all the participants or guardians prior to the study. All guardians were assured of the voluntary nature of the study and that all patient data were confidential. The study was approved by the Kandahar University Ethics Committee

(code number KDRU-EC-2020.59). For data collection, only patients' initials and medical registration numbers were used. Prior to entering into the computer for analysis, the collected data were coded and de-identified.

## Study design and participants

This hospital-based, prospective observational cohort study was conducted from February 2020–January 2021. Children were enrolled in the study if they from Kandahar province and were aged $< 18$ years with clinically suspected meningitis and had at least one of the following findings on examination of cerebrospinal fluid (CSF): (i) turbid appearance, (ii) leucocytosis >100 cells/mm3, and (iii) a leucocytosis of 10–100 cells/mm$^3$ with either an elevated protein level (>100 mg/dl) or decreased glucose concentration (<40 mg/dl) [15]).

Children were excluded from the study if they met any one of the exclusion criteria: (i) diagnosed with tuberculous meningitis (TBM), based on the definition by Marais et al [clinically suspected TBM plus a total diagnostic score of $\geq$10 points (cerebral imaging unavailable) or $\geq$12 points (cerebral imaging available) with at least 2 points from CSF or cerebral imaging criteria and exclusion of alternative diagnoses [16], (ii) clinically diagnosed viral meningitis: CSF analysis showing a lymphocytic predominance (>100 cells/mm$^3$), protein <45 mg/dl, and glucose >40 mg/dl), (iii) had received any parenteral broad-spectrum antibiotics before admission.

## Management of children with suspected ABM

All admitted children underwent a detailed history, physical examination and a lumbar puncture by our team of experienced paediatricians. Children with clinically diagnosed ABM were treated with combination of ceftriaxone (100 mg/kg/day) and ampicillin (150–200 mg/kg/day) for 10–14 days. Dexamethasone (0.6 mg/kg/day x 4 days) was administered to children aged $\geq$ 6 weeks at the physician's discretion. Children were discharged when afebrile and after completing their antibiotics. Follow up in the study was during the hospitalization days of the patients.

## Data collection & analysis

Data were collected on standardised case report forms (CRFs) and included socio-demographic characteristics, symptoms, physical signs, routine laboratory findings and CSF results. Vaccination status of the younger children was assessed by looking at the vaccination cards (or recall if the card was left at home) while, for the older children, it was generally by recall of the parents or caretakers. If the caretakers were unsure, other family members were contacted by phone.

Data were double entered into Microsoft Excel, and checked for accuracy before analysis using SPSS version 22 (Chicago, IL, USA). Frequency and percentage were used to summarise categorical variables. Chi-square test [using crude odd ratio (COR)] was performed to assess the association between two categorical variables. All variables that were statistically significant ($p$-value <0.05) in univariate analyses were assessed for independence in a binary logistic regression, using adjusted odds ratio (AOR), to determine the factors associated with death. Children with missing data for death were excluded from this analysis. A two-sided $p$-value of <0.05 was considered statistically significant.

## Results

Of the total of 718 patients were admitted with meningitis, 158 patients had TB meningitis, 121 encephalitis, 27 had received prehospital intravenous ceftriaxone, and 19 guardians

declined participation in the study. This left 393 patients who were enrolled in the study. Their mean age was 4.8 years [with 192 (48.8%) under age 5] and males accounted for just under 60% (Table 1). Some two thirds were rural dwellers, more than half [222 (56.5%)] came from families whose monthly income was <2,500 Afghanis (<30 USD), while 294 (74.8%) were living in households of >10 inhabitants. Only 96 (24.4%) children had been vaccinated for both Hib and PCV and only 94 (23.9%) had received the measles vaccine.

Main reported symptoms in the children included headache and seizures; while main signs were fever (body temperature of ≥38˚C), irritability, neck stiffness, Kernig sign, and coma. Evidence of concurrent infections were present in 250 (63.6%) of these patients, mainly tonsillitis/pharyngitis, pneumonia, and otitis media (Table 2).

The blood and CSF findings are shown in Table 3. The median CSF total white cell count was 110 cells/mm$^3$ with a range of 5 to 12,400 cells/mm$^3$; all children had a raised total protein but only 75.6% of children had a neutrophilic predominance.

All the patients were treated with combination of ceftriaxone and ampicillin, 169 (52.6%) received dexamethasone and the mean hospital stay was ~2 weeks. Among these children, 72 were discharged by their parents against medical advice and they could not be followed up. Most of the 72 patients were in critical situation, so possibly their parents thought they were dying. Consequently, a total of 321 children had a known outcome. Of these, 69 (21.5%) died and 96 (29.9%) had neurological sequelae, mostly focal muscle weakness and spasticity (Table 4).

Bivariate analysis identified coma on admission, no adjunctive dexamethasone therapy, no PCV or Hib vaccination, male gender and a purpuric/petechial rash as factors for death (Table 5); all except petechial rash were identified as independent variables for death in the logistic regression model (Table 6).

## Discussion

In this hospital-based, prospective observational cohort study, we have shown that a fifth of children with clinically diagnosed ABM died and a little under a third had neurological sequelae. The key risk factors we identified were coma on admission, not receiving dexamethasone, being male and lack of vaccination (PCV, Hib) against two common causes of meningitis whilst a purpuric/petechial rash was suggestive. The majority of our children were young (half < 5y), from poor families living in overcrowded conditions in rural areas.

The in-hospital case fatality was 21.5% in our study. The fact that 72 of the children were critical ill when brought home by their parents means that the mortality rate may have been much higher than the stated 21.5%. If all of these died, the mortality would have actually been 35.9%. Our death rate is higher compared to studies conducted in children from wealthier countries like Kosovo, 2.6% (age one month–16 years) [17], Iceland, 4.4% (age ≤18.5y) [18], South Korea, 9.5% (age ≤18y) [19], and Iran, 10% (5m–10y) [20]. One comparable study from India reported a 16% death rate in children aged one month–five years [21]. Higher mortality rates have been reported from Malawi, 28.7% (<15 years) [22], Nepal, 33.3% (≤15y) [23], Angola, 33% (2m–12y) [24], and Pakistan, 34% (<5 y) [25]. The differences in mortality rates observed in different studies are due to several reasons, including type of study, age spectrum of the children, definition of ABM, culture, and socio-economic, nutritional status and HIV status. Broader factors that are likely to be related to our high mortality include the fragile health system, a health infrastructure lacking basic facilities like microbiological diagnosis and intensive care for critically ill patients, the weak referral link between primary care and tertiary centres, and the lack of health education.

**Table 1. Baseline socio-demographic characteristics and vaccination data of ABM patients.**

| Variable | Number (Percentage) (N = 393) |
|---|---|
| Age (years), mean (SD) | 4.8 (3.5) |
| Age (years) | |
| <1 | 76 (19.3) |
| 1–5 | 116 (29.5) |
| 5–12 | 122 (31.1) |
| >12 | 79 (20.1) |
| Gender | |
| Male | 231 (58.8) |
| Female | 162 (41.2) |
| Number of siblings, mean (SD) | 6 (3) |
| Place of living | |
| Urban | 126 (32.1) |
| Rural | 267 (67.9) |
| Father's literacy level | |
| Illiterate | 264 (67.2) |
| Literate | 129 (32.8) |
| Primary | 102 |
| Secondary | 12 |
| Bachelor | 15 |
| Mother's literacy level | |
| Illiterate | 321 (81.7) |
| Literate | 72 (18.3) |
| Primary | 72 |
| Secondary | 0 |
| Bachelor | 0 |
| Father's occupation | |
| Self-employed | 177 (45.0) |
| Non-government employee | 168 (42.8) |
| Government employee | 24 (6.1) |
| Unemployed | 24 (6.1) |
| Family monthly income (in Afghanis) | |
| <2,500 (<30 USD) | 222 (56.5) |
| 2,500–20,000 (30–250 USD) | 159 (40.5) |
| >20,000 (>250 USD) | 12 (3.0) |
| Number of people living in the same house | |
| <5 | 3 (0.8) |
| 5–10 | 96 (24.4) |
| >10 | 294 (74.8) |
| Exclusive breastfeeding [a] | 378 (96.2) |
| PCV and Hib vaccination | 98 (24.9) |

[a] Feeding infants only breast milk during the first 6 months of life.

Hib, *Haemophilus influenzae* type b; PCV, pneumococcal vaccine; kg, SD, standard deviation; USD, United States Dollar.

**Table 2. Clinical features on admission in the 393 ABM patients.**

| Symptom/Sign | Number (Percentage) |
|---|---|
| | (N = 393) |
| Fever | 363 (92.4) |
| Vomiting | 321 (81.7) |
| Headache (N = 301) | 240 (79.7) |
| Anorexia | 312 (79.4) |
| Irritability | 297 (75.6) |
| Neck stiffness | 231 (58.8) |
| Coma [a] | 56 (14.2) |
| Seizures [b] | 372 (94.7) |
| Kernig's sign | 186 (47.3) |
| Brudzinski's sign | 15 (3.8) |
| Bulging fontanelle | 18 (4.6) |
| Purpuric/petechial rash | 21 (5.3) |
| Concurrent infections | 250 (63.6) |
| Type of concurrent infections (N = 250) [c] | |
| Tonsillitis/Pharyngitis | 108 (43.2) |
| Pneumonia | 72 (28.8) |
| Otitis media | 36 (14.4) |
| Measles | 12 (4.8) |
| Others | 22 (8.8) |

[a] Coma is defined as Glasgow Coma Scale (GCS) of $\leq 8$.

[b] Seizures observed on admission as well as history of seizures during 48 hours before hospitalisation.

[c] Among these 250 patients, 247 (98.8%) had single infections and 3 (1.2%) had double infections.

**Table 3. Full blood count and CSF findings in 393 ABM children.**

| Variable | Mean (SD) | Range |
|---|---|---|
| *Full blood count* | | |
| Haemoglobin (g/dl) | 10.0 (2.1) | 3.7–14.2 |
| Total leukocyte count (cells/mm$^3$) | 13,399 (7,862) | 1,700–35,000 |
| Neutrophils (%) | 64.4 (19.2) | 11.8–96.1 |
| Neutrophilia, N (%) | 296 (75.4) | |
| Lymphocytes (%) | 28.7 (16.0) | 2.3–66.0 |
| Platelets (cells/mm$^3$) | 321,350 (170,242) | 16,000–738,000 |
| *CSF findings* | | |
| Appearance | | |
| Clear | 194 | 49.4 |
| Turbid | 184 | 46.8 |
| Bloody | 15 | 3.8 |
| Total white blood cells (cells/mm$^3$), median (IQR) | 110 (95–128) | 5–12,400 |
| Neutrophils (%), median (IQR) | 70.8 (51.7–88.9) | |
| Neutrophilia (>50%), N (%) | 297 (75.6) | |
| Mononuclear cells (%), median (IQR) | 29.2 (11.1–48.3) | |
| Protein (mg/dl), mean (SD) | 182 (49) | 101–305 |
| Glucose (mg/dl), mean (SD) | 57.6 (23.8) | 8.0–111.0 |

CSF, cerebrospinal fluid; IQR, interquartile range; SD, standard deviation.

Categorical data are N (%).

**Table 4. Treatment and outcomes of 321 ABM children.**

| Variable (N = 321) | Number (N) | Percentage (%) |
|---|---|---|
| Admission in days | 13.6 (5.3) | |
| Days of antibiotics [a] | 12.9 ± 4.5 | |
| Type of complication (N = 96) [b] | | |
| Cerebral palsy | 26 | 27.1 |
| Hemiplegia | 23 | 23.9 |
| Hydrocephalus | 12 | 12.5 |
| Cranial nerve III palsy | 9 | 9.4 |
| Paraplegia | 8 | 8.3 |
| Aphasia | 7 | 7.3 |
| Deafness | 7 | 7.3 |
| Epilepsy | 4 | 4.2 |

[a] All ABM patients were treated with intravenous ceftriaxone and intravenous ampicillin.

[b] Among 96 patients, 94 (79.9%) had single complication, 2 (2.1%) had double, while none had >2 complications.

SD, standard deviation.

Many of our children had not received Hib and PCV vaccines, a public health failure. In Brazil, the introduction of pneumococcal and meningococcal vaccines in the childhood immunization programme was associated with a 50% nationwide decline in meningitis deaths in children aged 2-4y (1.46 to 0.72/100,000) and those < 2y (6.99 to 3.45/100,000) [26] and another Brazilian study showed that, in the under 2s, the pneumococcal vaccine reduced the incidence rates of pneumococcal meningitis by ~60% (6.01 to 2.49 cases/100,000) and deaths by 75% (1.85 to 0.47/100,000) [27]. Over the past three decades in the Netherlands, *H. influenzae* meningitis has declined from 1.57 to 0.14 per 100,000 population following the introduction of the Hib vaccine [28]. Clearly, PCV and Hib vaccines represent an excellent cost-effective strategy and, in high burden areas, will have incremental effects in reducing the burden of ABM [29].

Although a number of clinical trials have been conducted, adjuvant dexamethasone in the treatment of paediatric ABM still remains controversial. Mortality in our study was significantly lower in children who received adjunctive dexamethasone treatment but our study was not a randomised trial with matched groups. Most studies from high-income countries have demonstrated higher survival rates and improved overall outcomes following dexamethasone in the treatment of bacterial meningitis [30], such as studies from USA (all were children) [31, 32], France (all were children) [33], and Europe (all were adults) [34]. By contrast, several studies from low- and middle-income countries have not shown a clear benefit of dexamethasone [30], such as studies from Malawi (all were children with 24% HIV positive) [35], India (all were children with HIV status unknown) [36], Egypt (all were children HIV status unknown) [37], and Mozambique (HIV status unknown) [38]. A study from India reported that dexamethasone significantly reduced fatality in neonates [39]. Reasons for not using dexamethasone include anxiety regarding underlying HIV and the association of a higher mortality in patients with suspected ABM [40].

Surprisingly, in our study, a high percentage (25%) of mononuclear predominance was observed in CSF analysis. One of the possibilities could be that some of these patients were actually tuberculous meningitis.

**Table 5. Chi-square test of the factors associated with mortality in ABM patients.**

| Variable | Total | Survived | Died | COR | 95% CI | P-value |
|---|---|---|---|---|---|---|
| | N = 321 | N = 252 | N = 69 | | | |
| Age (years) | | | | | 0.4–1.1 | 0.076 |
| <5 | 156 (48.6) | 129 (82.7) | 27 (17.3) | 0.6 | | |
| ≥5 | 165 (51.4) | 123 (74.5) | 42 (25.5) | 1 | | |
| Gender | | | | | 1.3–4.2 | 0.004 |
| Male | 189 (58.9) | 138 (73.0) | 51 (27.0) | 2.3 | | |
| Female | 132 (41.1) | 114 (86.4) | 18 (13.6) | 1 | | |
| PCV and Hib vaccination | | | | | 1.4–6.7 | 0.004 |
| Yes | 80 (24.9) | 72 (90.0) | 8 (10.0) | 1 | | |
| No | 241 (75.1) | 180 (74.7) | 61 (25.3) | 3.0 | | |
| Place of living | | | | | 0.7–2.2 | 0.424 |
| Urban | 99 (30.8) | 75 (75.8) | 24 (24.2) | 1.3 | | |
| Rural | 222 (69.2) | 177 (79.7) | 45 (20.3) | 1 | | |
| Vomiting | | | | | 0.4–1.5 | 0.464 |
| Yes | 261 (81.3) | 207 (79.3) | 54 (20.7) | 0.8 | | |
| No | 60 (18.7) | 45 (75.0) | 15 (25.0) | 1 | | |
| Headache (N = 295) | | | | | 0.7–3.7 | 0.290 |
| Yes | 240 (82.4) | 195 (81.2) | 45 (18.8) | 1.5 | | |
| No | 55 (17.6) | 48 (87.3) | 7 (12.7) | 1 | | |
| Anorexia | | | | | 0.3–1.1 | 0.127 |
| Yes | 258 (80.4) | 207 (80.2) | 51 (19.8) | 0.6 | | |
| No | 63 (19.6) | 45 (71.4) | 18 (28.6) | 1 | | |
| Irritability | | | | | 0.2–0.8 | 0.005 |
| Yes | 72 (22.4) | 48 (66.7) | 24 (33.3) | 1 | | |
| No | 249 (77.6) | 204 (81.9) | 45 (18.1) | 0.4 | | |
| Neck stiffness | | | | | 0.5–1.5 | 0.576 |
| Yes | 177 (55.1) | 141 (79.7) | 36 (20.3) | 0.9 | | |
| No | 144 (44.9) | 111 (77.1) | 33 (22.9) | 1 | | |
| Coma on admission | | | | | 2.0–6.9 | <0.001 |
| Yes | 53 (16.5) | 30 (56.6) | 23 (43.4) | 3.7 | | |
| No | 268 (83.5) | 222 (82.8) | 46 (17.2) | 1 | | |
| Kernig sign | | | | | 0.7–1.9 | 0.702 |
| Yes | 147 (45.8) | 114 (77.6) | 33 (22.4) | 1.1 | | |
| No | 174 (54.2) | 138 (79.3) | 36 (20.7) | 1 | | |
| Purpuric/petechial rash | | | | | 1.5–10.6 | 0.002 |
| Yes | 18 (5.6) | 9 (50.0) | 9 (50.0) | 4.0 | | |
| No | 303 (94.4) | 243 (80.2) | 60 (19.8) | 1 | | |
| Concurrent infections | | | | | 0.5–1.5 | 0.638 |
| Yes | 211 (65.7) | 164 (77.7) | 47 (22.3) | 1 | | |
| No | 110 (34.3) | 88 (80.0) | 22 (20.0) | 0.9 | | |
| Dexamethasone therapy | | | | | 2.5–8.5 | <0.001 |
| Yes | 169 (52.6) | 152 (89.9) | 17 (10.1) | 1 | | |
| No | 152 (47.4) | 100 (65.8) | 52 (34.2) | 4.6 | | |

COR, Crude odds ratio; CSF, Cerebrospinal Fluid; Hib, *Haemophilus influenzae* type b; PCV, Pneumococcal Vaccine.

**Table 6. Independent factors associated with death in our children with ABM.**

| Risk factor | Category | COR (95%CI) | AOR (95%CI) | P-value |
|---|---|---|---|---|
| Dexamethasone therapy | Not given | 4.6 (2.5–8.5) | 4.9 (2.6–9.5) | <0.001 |
| Coma on admission | Present | 3.7 (2.0–6.9) | 4.6 (2.3–9.5) | <0.001 |
| PCV and Hib vaccination | Unvaccinated | 3.0 (1.4–6.7) | 2.8 (1.2–6.6) | 0.019 |
| Gender | Male | 2.3 (1.3–4.2) | 2.7 (1.4–5.5) | 0.005 |
| Purpuric/petechial rash | Present | 4.0 (1.5–10.6) | 3.1 (0.9–8.3) | 0.05 |

AOR, Adjusted odds ratio; COR, Crude odds ratio; Hib, *Haemophilus influenzae* type b; PCV, Pneumococcal Vaccine.

## Limitations

Although the main strength of our study was its prospective design in a real-life setting of a country ravaged by war and lacking significant resources, it had several limitations. We did not have essential microbiological diagnostics so none of our patients had a CSF Gram stain and bacteriologically proven meningitis. Just under a fifth of our patients discharged themselves before an outcome was known; therefore, our morbidity and mortality rates could be higher than reported. Although this was a single centre study conducted in a resourced limited referral hospital, our findings are probably applicable to most settings in Afghanistan and comparable settings in our region where HIV prevalence is low. Finally, we only assessed complications/sequelae at discharge and so cannot estimate long term morbidity and mortality and the effect on cognitive development and school performance.

## Conclusion

Presumed ABM in our setting was associated with high rates of morbidity and mortality. Increasing PCV and Hib vaccine coverage is likely to have a profound and positive effect on meningitis in Afghanistan. We found a beneficial effect of dexamethasone and this should be further investigated in countries in south Asia despite the discouraging data from Africa.

## Supporting information

**S1 Fig.**
(TIF)

**S1 Data.**
(SAV)

## Acknowledgments

We present our highest and sincere thanks to the authorities of Faculty of Medicine, Kandahar University and Kandahar Directorate of Public Health. We are also very thankful of the staff members (clinicians, nurses, lab technicians) of the hospital and all of our study participants.

## Author Contributions

**Conceptualization:** Bilal Ahmad Rahimi, Niamatullah Ishaq, Ghulam Mohayuddin Mudaser, Walter R. Taylor.

**Data curation:** Bilal Ahmad Rahimi.

**Formal analysis:** Bilal Ahmad Rahimi.

**Investigation:** Bilal Ahmad Rahimi, Niamatullah Ishaq, Ghulam Mohayuddin Mudaser.

**Methodology:** Bilal Ahmad Rahimi, Ghulam Mohayuddin Mudaser, Walter R. Taylor.

**Project administration:** Bilal Ahmad Rahimi, Walter R. Taylor.

**Resources:** Niamatullah Ishaq.

**Software:** Bilal Ahmad Rahimi.

**Supervision:** Bilal Ahmad Rahimi.

**Validation:** Bilal Ahmad Rahimi.

**Visualization:** Bilal Ahmad Rahimi.

**Writing – original draft:** Bilal Ahmad Rahimi.

**Writing – review & editing:** Niamatullah Ishaq, Ghulam Mohayuddin Mudaser, Walter R. Taylor.

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
