## [Editor Report · Decision Letter 0]

13 Jul 2021

PONE-D-21-20289

Treatment outcome of acute bacterial meningitis among children in Kandahar, Afghanistan: A prospective study.

PLOS ONE

Dear Dr. Rahimi,

Thank you for submitting your manuscript to PLOS ONE. After careful consideration, we feel that it has merit but does not fully meet PLOS ONE’s publication criteria as it currently stands. Therefore, we invite you to submit a revised version of the manuscript that addresses the points raised during the review process.

Note that I am send this request for revision before sending the manuscript out for review in that there are things that must be addressed before a review can be done. The first and most important is that a definition of acute bacterial meningitis must be provided. Second, there is a description of CSF findings but only with a range given that includes WBCs as low as 5 and neutrophil counts as low as 0 or lymphocytes up to 100. What is the reason for believing that these were bacterial (or were not TB)? It may still be difficult to publish in view of the lack of microbiologic data but at least this will improve the chance. On another note, the sample size calculation appears to be irrelevant to the study being reported.

We look forward to receiving your revised manuscript.

Kind regards,

Rodney D Adam

Academic Editor

PLOS ONE
---

## [Author Response · Author response to Decision Letter 0]

17 Jul 2021

Hello dear academic Editor,

Many thanks for your prompt reply and useful comments. These are really important points.

Following are your comments and our answers:

Comment 1. The first and most important is that a definition of acute bacterial meningitis must be provided.

Answer: We had written the definition at the end of “Material and Methods”. Our definition is based on WHO guidelines. Now we gave it a heading, and is as follows:

Definition of ABM

ABM was defined according to the guidelines of WHO, i.e., a CSF examination showing at least one of the following characteristics: turbid CSF appearance, leukocytosis >100 cells/mm3, and leukocytosis of 10–100 cells/mm3 with either an elevated protein level (>100 mg/dl) or decreased glucose level (<40 mg/dl) [14].

Comment 2. Second, there is a description of CSF findings but only with a range given that includes WBCs as low as 5 and neutrophil counts as low as 0 or lymphocytes up to 100. What is the reason for believing that these were bacterial (or were not TB)?

Answer: We have selected and recruited all the ABM patients based on CSF analysis results from the laboratory. Lab results were analyzed based on the definition by WHO, as mentioned in the answer of first comment. We recruited the patient if he/she had any one of the following 3 findings in CSF analysis:

• Turbid CSF appearance.

• Leukocytosis >100 cells/mm3.

• Leukocytosis of 10–100 cells/mm3 with either an elevated protein level (>100 mg/dl) or decreased glucose level (<40 mg/dl).

Comment 3. One another note, the sample size calculation appears to be irrelevant to the study being reported.

Answer: Yes, we agree with you. It is irrelevant here. So, we have corrected the sample size calculation in the manuscript as follows:

Sample size calculations

All eligible patients, who were admitted to pediatric ward of Mirwais regional hospital during 12-month-period (February 2020–January 2021) and willing to take part in the study, were included in this study. So, there was no need for the sample size calculation.

---

## [Decision Letter · Decision Letter 1]

9 Sep 2021

PONE-D-21-20289R1Treatment outcome of acute bacterial meningitis among children in Kandahar, Afghanistan: A prospective study.PLOS ONE

Dear Dr. Rahimi,

Thank you for submitting your manuscript to PLOS ONE. After careful consideration, we feel that it has merit but does not fully meet PLOS ONE’s publication criteria as it currently stands. Therefore, we invite you to submit a revised version of the manuscript that addresses the points raised during the review process.

Please do a careful response to the reviews and include your point by point response to the items raised. Please submit your revised manuscript by Oct 24 2021 11:59PM. If you will need more time than this to complete your revisions, please reply to this message or contact the journal office at plosone@plos.org. Please include the following items when submitting your revised manuscript:A rebuttal letter that responds to each point raised by the academic editor and reviewer(s). You should upload this letter as a separate file labeled 'Response to Reviewers'.A marked-up copy of your manuscript that highlights changes made to the original version. You should upload this as a separate file labeled 'Revised Manuscript with Track Changes'.An unmarked version of your revised paper without tracked changes. You should upload this as a separate file labeled 'Manuscript'.

We look forward to receiving your revised manuscript.

Kind regards,

Rodney D Adam

Academic Editor

PLOS ONE

Journal Requirements:

Additional Editor Comments (if provided):

Reviewers' comments:

Reviewer's Responses to Questions

**Comments to the Author**

1. If the authors have adequately addressed your comments raised in a previous round of review and you feel that this manuscript is now acceptable for publication, you may indicate that here to bypass the “Comments to the Author” section, enter your conflict of interest statement in the “Confidential to Editor” section, and submit your "Accept" recommendation.

Reviewer #1: (No Response)

Reviewer #2: (No Response)

2. Is the manuscript technically sound, and do the data support the conclusions?

Reviewer #1: Partly

Reviewer #2: No

3. Has the statistical analysis been performed appropriately and rigorously? 

Reviewer #1: No

Reviewer #2: No

4. Have the authors made all data underlying the findings in their manuscript fully available?

Reviewer #1: Yes

Reviewer #2: Yes

5. Is the manuscript presented in an intelligible fashion and written in standard English?

Reviewer #1: Yes

Reviewer #2: Yes

6. Review Comments to the Author

Reviewer #1: Abstract

Line 25: Background – no need to say handicap as this is covered by the term morbidity

Introduction

- Reference number 3 is not useful to this paper as it is from 1993 and treatment and outcomes have improved much since then

- Line 59: remove “three most common bacteria”

- Lines 59 – 61: the information included from reference no. 4 is outdated and has been replaced with a newer Lancet study (GBD 2016 Meningitis Collaborators. Global, regional, and national burden of meningitis, 1990-2016: a systematic analysis for the Global Burden of Disease Study 2016. Lancet Neurol. 2018 Dec;17(12):1061-1082)

- Consider including some data from the region (as local data is not available)

Methodology

- Lines 83 – 85: need to indicate which data was collected from the questionnaire and to whom were these questionnaires administered. Also, indicate which data was abstracted from patient files.

- How was vaccination status assessed?

- Line 95: define what treatment outcomes your study is looking at

- Line 96: describe the factors that you will include in assessing this objective

- Line 99: who made the diagnosis of ABM and how was it made i.e. primary clinician Or study investigator? If made by clinician please provide some detail on how the clinicians in the facility make this diagnosis? is made in this facility). If by study investigator, explain at what point and how this was done.

- Lines 103 & 104: how were the diagnoses of TBM and encephalitis made and confirmed; Note that CSF findings from table 5 suggest that patients might have had TBM and viral meningitis/encephalitis

- Line 105: why were patients with a history of parenteral broad-spectrum antibiotics use before admission excluded from the study? Also define which antibiotics were included in this category

- Line 110: instead of sample size calculation, power calculation can be done and they can be used to demonstrate the minimum detectable difference with different sample sizes.

- Line 128 – 133: The definition of ABM – not clear why this has been included – this should be covered under line 99 and has been addressed above.

Results:

- Consider providing the flow of study participants – i.e. how many patients were excluded by each criteria indicated in lines 103 - 106

- The results of variables with outliers should include median (IQR) e.g. age, duration, CSF cell count, differentials etc.

- The overall presentation of results in tables 1 and 6 should be done in a logical and more meaningful manner. Consider reducing the number of tables and assign variables to the appropriate table i.e.

o Table 1 : baseline demographic and social characteristics including vaccination data (can have both categorical and continuous data)

o Table 2: clinical presentation

o Table 3: lab findings including CSF findings

o Table 4: treatment and outcome findings

- There is no need to include the total row for each variable under all tables (a single N at the top should suffice) and also there is no need to include the total number in the write-up of the results, just the number and percent suffices e.g. no need for 138/339 (58.8%) for male gender.

- De-clutter the tables – there are too many variables being presented and not all are necessary; consider dropping e.g. number in the same room, no. of <5 in the same room, age of starting breastfeeding, birth order, twins, mode of delivery etc

- Dexamethasone therapy, Duration of admission (LOS), no of days of iv ceftriaxone and no. of days of iv ampicillin should be in the treatment & outcome table (not in table 1)

- Also consider presenting antibiotic data as one variable i.e. duration of anti-meningitic antibiotics

- Some variables presented in the results are not clear and should be defined either in the methodology section or as a footnote in the tables e.g. income categories, breastfeeding, bottle feeding;

- The symptoms and signs variables should also be defined in terms of, at what point were they obtained i.e. at admission, or at any point in their stay

- Age categories – consider having an additional category for above 12 years

- Weight and height should be presented as wt for age and ht for age

- Vaccination variables: Why have the HiB and PCV vaccinations been combined into a single variable?

- Comorbidity variable: This variable needs to be defined and categorized as either acute or chronic; the acute conditions are most likely concurrent infections; need to reconsider how this data has been presented. Moreover, the acute concurrent infections should be in the clinical presentations table while the chronic underlying co-morbidities should be in the baseline characteristics table

- Table with lab results should include units e.g. TLC, platelets, WBC in CSF

- Results write-up should include the most relevant information from the tables and to the overall objective e.g.

o Lines 156-158 should include something on next stiffness or headaches

o Lines 162-163: should include information on CSF white cell count and differential results

- Consider excluding the patients that were self-discharged from the treatment & outcome table as they could not be assessed for complications and outcomes

- Table 7:

o Current table 7 analysis is testing for factors associated with survival not mortality: redo this table and assess for mortality across all variables and should indicate the reference categories against each variable, as this has also not been done in a uniform manner

o The values presented for the OR and p-value for living in over-crowded areas is incorrect, needs to be redone; also reconsider using this variable as 99.2% of participants were living in the crowded areas (>5).

o This table does not need the total number in each row as well (total N at the top is good enough)

o De-clutter this table similar to the suggestion for table 1 and 2 above

o Consider not testing for complications as a factor associated with mortality. Most complications assessed in this study are likely assessed at discharge or at recovery. Also, death can be considered as a complication of ABM.

o Assess acute and chronic co-morbidities separately

Discussion

- Case Fatality Rate – exclude the number who were self-discharged

- The references given for the proportions of case fatality and complication from the various countries should also include the populations in which these studies were done as they are not all the same e.g. some are in infants, others in children under 5 and others across all pediatric age groups

- Lines 240 – Should read: …will have incremental effects in reducing the global burden…

- As mentioned above, living in crowded areas has been incorrectly included

Conclusion:

- Line 274: incorrect conclusion: your study does not prove the cost-effectiveness of vaccination

- Lines 275-276: not sure how the assessment of outcomes and factors was hindered by unavailability of culture confirmed cases

Reviewer #2: August 28, 2021

PLOS ONE

Manuscript ID: PONE-D-21-20289R1

Title: Treatment outcome of acute bacterial meningitis among children in Kandahar, Afghanistan: A prospective study

Thank you for the opportunity to review this interesting manuscript. The authors conducted a prospective observational cohort study of 393 children <18 years of age with acute bacterial meningitis (ABM) at Mirwais Regional Hospital in Afghanistan from February 2020 through January 2021. The sociodemographic components reflected many patients coming from low-income families living in overcrowded households in rural areas and with incomplete childhood vaccination programmes. The in-hospital mortality was 17.6% and adjusted analyses suggested not having received vaccination for Hib or PCV, living in overcrowded houses, complications during admission, and lack of adjunctive dexamethasone as poor prognostic factors.

The authors should be commended for performing an important study during very difficult and challenging circumstances. The scientific community definitely needs more data from developing countries. However, the methodology can still be improved in several ways which I believe would strengthen the impact and credibility of the data substantially.

Major concerns:

1. Abstract. Conclusion. Although there is no doubt that Hib and PCV vaccinations prevent ABM, the provided data do not document a protective effect of Hib or PCV vaccinations on incidence of ABM in children in Afghanistan. This would require a comparison of risk of ABM among vaccinated and unvaccinated children. So, the conclusion that their findings prove the value does not seem to be justified in this dataset. However, they can/should still encourage more widespread implementation of the vaccinations, which makes sense. Please comment.

2. It is my overall impression from the provided methods and the discussion, that they examine in-hospital mortality as primary outcome, which should be clearly specified throughout the abstract and manuscript. Of note, this is not the same as ‘case-fatality rate’ which is also used once in the discussion. Please correct if I understand your study correctly.

3. ‘Comorbidities’ usually denote pre-existing conditions such as diabetes mellitus, arterial hypertension etc. Suggest changing to ‘predisposing’ or ‘concomitant’ infection throughout the manuscript.

4. The authors conducted a prospective observational cohort study. How were the data retrieved, personal interview or medical record review?

5. There are no missing values at all, which seems almost impossible. Especially since some patients were up to 15 years old and the parents may not remember for how long the child was breastfed, length of bottle-feeding etc (which is also likely to be irrelevant in adolescents). Another example is presence of headache which is 100% complete but many cases were babies who are very unlikely to be able to account for presence of headache or not. Were families and relatives contacted to complete all variables and what if they couldn’t be reached? Please comment.

6. Please specify how you categorized socio-economic status and income into e.g. low, middle, and high-income families?

7. Please describe who retrieved the study data or performed the interviews (single or multiple authors)?

8. Table 4. Is it symptoms and signs at admission or in total during course of illness? The proportion with seizures seems extremely high (95%). History of seizures or observed? Misclassification? Please comment?

9. Co-infections and complications, e.g. in Table 6. Some patients were likely to have had several co-infections (e.g. otitis media and pneumonia) or complications (seizures and cerebral palsy). Please account for this in the data provided.

10. Table 7. Please show the reference used for each crude (and adjusted) odds ratio, i.e. male sex in analyses of sex as prognostic factor etc.

Minor suggestions:

1. Title. Suggest adding ‘cohort study’ to ‘a prospective study’.

2. Abstract. Suggest specifying ‘complications’.

3. Abstract. The results suggest ‘not administering Hib and PCV vaccines’ was a poor prognostic factor. Would it not be more accurate to write ‘no Hib or PCV vaccines’ since I assume that they refer to children receiving childhood vaccinations before falling ill?

4. Please clarify in the methods which diagnostic methods were available in your setting, e.g. microscopy of CSF but no culture etc.

5. Suggest to simply put in ‘Not applicable’ under sample size calculations.

6. Line 141-143 is a bit unclear to me. Please clarify.

7. Please define ‘coma’ in your study.

8. Suggest to also include differences in study populations and definitions of ABM as a reason for differences in mortality across studies.

9. Line 177 onwards you should delete “%” after the 95% confidence interval and the p-value does not add any information, e.g.: “crude odds ratio [COR] 0.1, 95% CI 0.0-0.1%, and p-value <0.001).

10. The crude odds ratio for coma for in-hospital mortality is currently 0.1 (95% CI 0.0-0.1), which is clearly reversed?

11. Please also check that you have the right reference level in adjusted odds ratios in Table 8, e.g. for adjunctive dexamethasone currently reads as an AOR of 6.2 for in-hospital mortality?

12. Line 234-237. The sentences are a bit unclear with different numbers?

13. Line 242. Overcrowded houses are likely a risk factor for contracting ABM, but your study examines the prognosis after falling ill with ABM. However, it may be a proxy for poverty and a general poor health and thereby an adverse prognostic factor?

14. Line 275. This should be moved to limitations instead of in the conclusions.

Best of luck to the authors.

7. PLOS authors have the option to publish the peer review history of their article (what does this mean?). If published, this will include your full peer review and any attached files.

Reviewer #1: No

Reviewer #2: **Yes: **Jacob Bodilsen

---

## [Author Response · Author response to Decision Letter 1]

18 Sep 2021

First of all, I am very thankful of the useful comments of the two reviewers. I really appreciate it. As we have very less facilities (especially in research) in war-torn Afghanistan, we have tried our best to show the world a real picture of pediatric acute bacterial meningitis in Kandahar, Afghanistan. But I agree that there were many mistakes in our manuscript. Thanks again for your kind and fruitful comments.

6. Review Comments to the Author

Reviewer #1: Abstract

Line 25: Background – no need to say handicap as this is covered by the term morbidity

I removed it based on your comment.

Introduction

- Reference number 3 is not useful to this paper as it is from 1993 and treatment and outcomes have improved much since then

I removed the old reference. Now the references are up-to-date.

- Line 59: remove “three most common bacteria”

I removed it.

- Lines 59 – 61: the information included from reference no. 4 is outdated and has been replaced with a newer Lancet study (GBD 2016 Meningitis Collaborators. Global, regional, and national burden of meningitis, 1990-2016: a systematic analysis for the Global Burden of Disease Study 2016. Lancet Neurol. 2018 Dec;17(12):1061-1082)

Thanks for the link. I removed old data. Now the data is replaced with the reference that you had mentioned.

- Consider including some data from the region (as local data is not available)

I added the data of Afghanistan and its surrounding neighbor countries.

Methodology

- Lines 83 – 85: need to indicate which data was collected from the questionnaire and to whom were these questionnaires administered. Also, indicate which data was abstracted from patient files.

I added the following details:

Data was collected during 12-month-period (February 2020–January 2021) using questionnaire with questions regarding socio-demographic characteristics, physical signs and symptoms, laboratory examinations, and treatment of the patients admitted with the diagnosis of ABM. All the patients were followed up during the whole period of their hospitalization. All the questionnaires were filled by the study investigator. For all the patients, detailed history and physical examinations was conducted by experienced pediatricians while laboratory investigations and treatment data were abstracted from the patient files.

- How was vaccination status assessed?

I added these details:

Vaccination status of the younger children were assessed by looking at the vaccination cards (or recall if the card was left at home) while for the older children by recall of the parents or caretakers. If the caretakers did not have more and accurate information, other family members of the patients were contacted via phone calls better and accurate data collection.

- Line 95: define what treatment outcomes your study is looking at

To assess the treatment outcome (survived or died) of ABM among children in Kandahar province, Afghanistan.

- Line 96: describe the factors that you will include in assessing this objective

To determine the risk factors (including socio-demographic status, physical signs and symptoms, laboratory examinations, and treatment) associated with death among ABM patients.

- Line 99: who made the diagnosis of ABM and how was it made i.e. primary clinician Or study investigator? If made by clinician please provide some detail on how the clinicians in the facility make this diagnosis? is made in this facility). If by study investigator, explain at what point and how this was done.

Patient admitted in pediatric ward with the diagnosis of ABM (diagnosis was made by expert pediatricians of the pediatric ward, based on WHO guidelines, i.e., a CSF examination showing at least one of the following characteristics: turbid CSF appearance, leukocytosis >100 cells/mm3, and leukocytosis of 10–100 cells/mm3 with either an elevated protein level (>100 mg/dl) or decreased glucose level (<40 mg/dl) [14]).

- Lines 103 & 104: how were the diagnoses of TBM and encephalitis made and confirmed; Note that CSF findings from table 5 suggest that patients might have had TBM and viral meningitis/encephalitis

• Tuberculous meningitis patients (Clinical entry criteria plus a total diagnostic score of ≥10 points (when cerebral imaging is not available) or ≥12 points (when cerebral imaging is available) plus, exclusion of alternative diagnoses. At least 2 points should either come from CSF or cerebral imaging criteria) [15].

• Viral meningitis patients (patients with clinical diagnosis of viral meningitis and CSF analysis showing lymphocytic predominance [>100 cells/mm3], protein <45 mg/dl, and glucose >40 mg/dl).

- Line 105: why were patients with a history of parenteral broad-spectrum antibiotics use before admission excluded from the study? Also define which antibiotics were included in this category.

I added these lines:

Children who received any parenteral broad-spectrum antibiotics before admission (as these antibiotics may interfere with the results of CSF analysis). The two groups of broad-spectrum antibiotics used and available in Kandahar are third generation cephalosporins and macrolides.

- Line 110: instead of sample size calculation, power calculation can be done and they can be used to demonstrate the minimum detectable difference with different sample sizes.

The second reviewer also commented on the sample size calculation. He commented to write “Not applicable” in it. So, I did that. Hope you also agree with that.

- Line 128 – 133: The definition of ABM – not clear why this has been included – this should be covered under line 99 and has been addressed above.

OK. I addressed it under line 99, as per your comment.

Results:

- Consider providing the flow of study participants – i.e. how many patients were excluded by each criteria indicated in lines 103 – 106

Based on your comment, I added the following information in the result section:

Among the excluded patients, 158 patients had TB meningitis, 121 had encephalitis, 27 had received intravenous ceftriaxone before admission, while parents of 19 patients refused to participate in the study. 

- The results of variables with outliers should include median (IQR) e.g. age, duration, CSF cell count, differentials etc.

I added median (IQR) for all the variables with outliers.

- The overall presentation of results in tables 1 and 6 should be done in a logical and more meaningful manner. Consider reducing the number of tables and assign variables to the appropriate table i.e.

o Table 1 : baseline demographic and social characteristics including vaccination data (can have both categorical and continuous data)

o Table 2: clinical presentation

o Table 3: lab findings including CSF findings

o Table 4: treatment and outcome findings

Based on your comment, I reduced the number of tables from 6 to 4. Now all the relevant data is in its appropriate table, both categorical and continuous.

- There is no need to include the total row for each variable under all tables (a single N at the top should suffice) and also there is no need to include the total number in the write-up of the results, just the number and percent suffices e.g. no need for 138/339 (58.8%) for male gender.

OK. I removed the total row from the variables in all the tables. A single (e.g., N=393) has been added at the top titles row. Also I removed the total number in the results.

- De-clutter the tables – there are too many variables being presented and not all are necessary; consider dropping e.g. number in the same room, no. of <5 in the same room, age of starting breastfeeding, birth order, twins, mode of delivery etc

I removed all the unnecessary variables from the tables.

- Dexamethasone therapy, Duration of admission (LOS), no of days of iv ceftriaxone and no. of days of iv ampicillin should be in the treatment & outcome table (not in table 1)

I transferred the data to the treatment and outcome table.

- Also consider presenting antibiotic data as one variable i.e. duration of anti-meningitic antibiotics.

OK. I changed them to one variable, i.e., duration of anti-meningitis antibiotics.

- Some variables presented in the results are not clear and should be defined either in the methodology section or as a footnote in the tables e.g. income categories, breastfeeding, bottle feeding;

OK. I defined them.

- The symptoms and signs variables should also be defined in terms of, at what point were they obtained i.e. at admission, or at any point in their stay.

OK. These symptoms were at the time of admission.

- Age categories – consider having an additional category for above 12 years

OK. I added another category of >12 years of age.

- Weight and height should be presented as wt for age and ht for age

- Vaccination variables: Why have the HiB and PCV vaccinations been combined into a single variable?

I had combined the Hib and PCV vaccines because they both are administered at the same age and time (on week 6, 10 and 14). 

But now as per your comment, I separated them as 2 separate variables.

- Comorbidity variable: This variable needs to be defined and categorized as either acute or chronic; the acute conditions are most likely concurrent infections; need to reconsider how this data has been presented. Moreover, the acute concurrent infections should be in the clinical presentations table while the chronic underlying co-morbidities should be in the baseline characteristics table

OK. I corrected this variable. Now its name is changed to “concurrent infections”. Also, I put it in clinical presentation table.

- Table with lab results should include units e.g. TLC, platelets, WBC in CSF

OK. Now they all have the units, e.g., cells/mm3.

- Results write-up should include the most relevant information from the tables and to the overall objective e.g.

o Lines 156-158 should include something on next stiffness or headaches

o Lines 162-163: should include information on CSF white cell count and differential results

OK. I added the data of the variables that you mentioned.

- Consider excluding the patients that were self-discharged from the treatment & outcome table as they could not be assessed for complications and outcomes.

OK. Thanks. It was really a big mistake. Now we corrected it. Self-discharged patients are now removed from the outcome analysis. I re-do all the analysis based on you comment.

- Table 7:

o Current table 7 analysis is testing for factors associated with survival not mortality: redo this table and assess for mortality across all variables and should indicate the reference categories against each variable, as this has also not been done in a uniform manner

I did the analysis again in all the variables. I also added the reference categories.

o The values presented for the OR and p-value for living in over-crowded areas is incorrect, needs to be redone; also reconsider using this variable as 99.2% of participants were living in the crowded areas (>5).

OK. I removed the variable of “living in over-crowded areas” from the analysis.

o This table does not need the total number in each row as well (total N at the top is good enough).

OK. I removed all the “total” rows from each variable and put N at the top.

o De-clutter this table similar to the suggestion for table 1 and 2 above.

OK. I did it.

o Consider not testing for complications as a factor associated with mortality. Most complications assessed in this study are likely assessed at discharge or at recovery. Also, death can be considered as a complication of ABM.

OK. Thanks. It was really a mistake. I removed the variable of “complications” from the analysis.

o Assess acute and chronic co-morbidities separately

OK. I changed it as per your previous comment and changed it to “concurrent infections”.

Discussion

- Case Fatality Rate – exclude the number who were self-discharged.

OK. I removed the self-discharge patients from this.

- The references given for the proportions of case fatality and complication from the various countries should also include the populations in which these studies were done as they are not all the same e.g. some are in infants, others in children under 5 and others across all pediatric age groups.

OK. It is done. Now I included the information of study participants with data of each country.

- Lines 240 – Should read: …will have incremental effects in reducing the global burden…

OK. I corrected and added it.

- As mentioned above, living in crowded areas has been incorrectly included.

OK. I have removed the living in over-crowded areas from the analysis.

Conclusion:

- Line 274: incorrect conclusion: your study does not prove the cost-effectiveness of vaccination.

It was a mistake. Now we corrected it.

- Lines 275-276: not sure how the assessment of outcomes and factors was hindered by unavailability of culture confirmed cases.

OK. We removed this sentence. Our aim by this sentence was the unavailability of definite diagnosis.

Reviewer #2: August 28, 2021

PLOS ONE

Manuscript ID: PONE-D-21-20289R1

Title: Treatment outcome of acute bacterial meningitis among children in Kandahar, Afghanistan: A prospective study

Thank you for the opportunity to review this interesting manuscript. The authors conducted a prospective observational cohort study of 393 children <18 years of age with acute bacterial meningitis (ABM) at Mirwais Regional Hospital in Afghanistan from February 2020 through January 2021. The sociodemographic components reflected many patients coming from low-income families living in overcrowded households in rural areas and with incomplete childhood vaccination programmes. The in-hospital mortality was 17.6% and adjusted analyses suggested not having received vaccination for Hib or PCV, living in overcrowded houses, complications during admission, and lack of adjunctive dexamethasone as poor prognostic factors.

The authors should be commended for performing an important study during very difficult and challenging circumstances. The scientific community definitely needs more data from developing countries. However, the methodology can still be improved in several ways which I believe would strengthen the impact and credibility of the data substantially.

Major concerns:

1. Abstract. Conclusion. Although there is no doubt that Hib and PCV vaccinations prevent ABM, the provided data do not document a protective effect of Hib or PCV vaccinations on incidence of ABM in children in Afghanistan. This would require a comparison of risk of ABM among vaccinated and unvaccinated children. So, the conclusion that their findings prove the value does not seem to be justified in this dataset. However, they can/should still encourage more widespread implementation of the vaccinations, which makes sense. Please comment.

Thanks for the comment. Now I corrected the sentence with encouraging more widespread implementation of the Hib and PCV vaccinations.

2. It is my overall impression from the provided methods and the discussion, that they examine in-hospital mortality as primary outcome, which should be clearly specified throughout the abstract and manuscript. Of note, this is not the same as ‘case-fatality rate’ which is also used once in the discussion. Please correct if I understand your study correctly.

OK. It was our mistake. Now it is changed to “in-hospital mortality”.

3. ‘Comorbidities’ usually denote pre-existing conditions such as diabetes mellitus, arterial hypertension etc. Suggest changing to ‘predisposing’ or ‘concomitant’ infection throughout the manuscript.

OK. This comment was given by the other respectable reviewer also. I corrected it and changed to “concurrent infections”.

4. The authors conducted a prospective observational cohort study. How were the data retrieved, personal interview or medical record review?

All the questionnaires were filled by the study investigators. For all the patients, detailed history and physical examinations was conducted by experienced pediatricians while laboratory investigations and treatment data were abstracted from the patient files.

5. There are no missing values at all, which seems almost impossible. Especially since some patients were up to 15 years old and the parents may not remember for how long the child was breastfed, length of bottle-feeding etc (which is also likely to be irrelevant in adolescents). Another example is presence of headache which is 100% complete but many cases were babies who are very unlikely to be able to account for presence of headache or not. Were families and relatives contacted to complete all variables and what if they couldn’t be reached? Please comment.

Yes, we agree with you. It is very difficult not to have missing data. Although we have very limited resources here in Afghanistan, but I am very fortunate to have a very committed team. They tried a lot not to miss any information. But I am sure that there might be recall bias in the answers of mothers or caretakers of the children. For breastfeeding, it was information of exclusive breastfeeding (only breastfeeding during first 6 months of life). For headache, we included we had included the very young children as “No headache”. Now we corrected it by putting headache as “not applicable” or “missing” data.

Vaccination status of the younger children were assessed by looking at the vaccination cards (or recall if the card was left at home) while for the older children by recall of the parents or caretakers. If the caretakers did not have more and accurate information, other family members of the patients were contacted via phone calls better and accurate data collection.

6. Please specify how you categorized socio-economic status and income into e.g. low, middle, and high-income families?

We had put it based on monthly income of the family. Now we changed SES to “monthly income”.

7. Please describe who retrieved the study data or performed the interviews (single or multiple authors)?

All the questionnaires were filled by the study investigators (we had 3 data collectors). For all the patients, detailed history and physical examinations was conducted by experienced pediatricians while laboratory investigations and treatment data were abstracted from the patient files.

8. Table 4. Is it symptoms and signs at admission or in total during course of illness? The proportion with seizures seems extremely high (95%). History of seizures or observed? Misclassification? Please comment?

The signs and symptoms were on admission. Now we added “on admission” with clinical features. Regarding seizures, it was the presence of seizures in the last 48 hours before admission.

9. Co-infections and complications, e.g. in Table 6. Some patients were likely to have had several co-infections (e.g. otitis media and pneumonia) or complications (seizures and cerebral palsy). Please account for this in the data provided.

Yes, you are absolutely right. We should have added this data. Now we added the data of patients who has more than one concurrent infections or complications.

10. Table 7. Please show the reference used for each crude (and adjusted) odds ratio, i.e. male sex in analyses of sex as prognostic factor etc.

OK. Now we added the references.

Minor suggestions:

1. Title. Suggest adding ‘cohort study’ to ‘a prospective study’

OK. Now “A prospective observational cohort study” is added to the title.

2. Abstract. Suggest specifying ‘complications’.

OK. Now we specified the main complications in the abstract.

3. Abstract. The results suggest ‘not administering Hib and PCV vaccines’ was a poor prognostic factor. Would it not be more accurate to write ‘no Hib or PCV vaccines’ since I assume that they refer to children receiving childhood vaccinations before falling ill?

OK. We corrected it as ‘no Hib or PCV vaccines’ both in abstract in the main text.

4. Please clarify in the methods which diagnostic methods were available in your setting, e.g. microscopy of CSF but no culture etc.

OK. Now we clarified this important point in the manuscript. 

5. Suggest to simply put in ‘Not applicable’ under sample size calculations.

OK. I made it as per your comment.

6. Line 141-143 is a bit unclear to me. Please clarify.

OK. Now the sentence is made clear.

7. Please define ‘coma’ in your study.

OK. Now definition of comma is added. “Coma is defined as a state of deep unconsciousness from which the affected person cannot be aroused even by strong stimulation.”

8. Suggest to also include differences in study populations and definitions of ABM as a reason for differences in mortality across studies.

OK. Now we have added them.

9. Line 177 onwards you should delete “%” after the 95% confidence interval and the p-value does not add any information, e.g.: “crude odds ratio [COR] 0.1, 95% CI 0.0-0.1%, and p-value <0.001).

10. The crude odds ratio for coma for in-hospital mortality is currently 0.1 (95% CI 0.0-0.1), which is clearly reversed?

OK. % and p-values have been removed from these places. We did the analysis again. You were right. Now COR of coma is 3.7.

11. Please also check that you have the right reference level in adjusted odds ratios in Table 8, e.g. for adjunctive dexamethasone currently reads as an AOR of 6.2 for in-hospital mortality?

OK. We again did the analysis of all the variables. 

12. Line 234-237. The sentences are a bit unclear with different numbers?

OK. Those 2 sentences were unclear. Now it is OK and clear.

13. Line 242. Overcrowded houses are likely a risk factor for contracting ABM, but your study examines the prognosis after falling ill with ABM. However, it may be a proxy for poverty and a general poor health and thereby an adverse prognostic factor?

OK. The other reviewer had also mentioned this issue and suggest to remove it as nearly all (99.2%) of participants were living in the over-crowded areas. So I removed this variable.

14. Line 275. This should be moved to limitations instead of in the conclusions.

OK. I moved this sentence to the limitations of the study.

Best of luck to the authors.

Many thanks for your kind comments.

---

## [Decision Letter · Decision Letter 2]

25 Oct 2021

PONE-D-21-20289R2Treatment outcome of acute bacterial meningitis among children in Kandahar, Afghanistan: A prospective study.PLOS ONE

Dear Dr. Rahimi,

Thank you for submitting your manuscript to PLOS ONE. After careful consideration, we feel that it has merit but does not fully meet PLOS ONE’s publication criteria as it currently stands. Therefore, we invite you to submit a revised version of the manuscript that addresses the points raised during the review process.

All the reviewer comments should be addressed in your revision.

We look forward to receiving your revised manuscript.

Kind regards,

Rodney D Adam

Academic Editor

PLOS ONE

Journal Requirements:

Reviewers' comments:

Reviewer's Responses to Questions

**Comments to the Author**

1. If the authors have adequately addressed your comments raised in a previous round of review and you feel that this manuscript is now acceptable for publication, you may indicate that here to bypass the “Comments to the Author” section, enter your conflict of interest statement in the “Confidential to Editor” section, and submit your "Accept" recommendation.

Reviewer #1: (No Response)

2. Is the manuscript technically sound, and do the data support the conclusions?

Reviewer #1: Partly

3. Has the statistical analysis been performed appropriately and rigorously? 

Reviewer #1: No

4. Have the authors made all data underlying the findings in their manuscript fully available?

Reviewer #1: Yes

5. Is the manuscript presented in an intelligible fashion and written in standard English?

Reviewer #1: Yes

6. Review Comments to the Author

Reviewer #1: Overall ,the authors have done a great job in improving their manuscript. They must be commended for taking their time in addressing each and every single issue that had been raised and have done so with great detail.

Nevertheless, there are still some issues that need to be addressed and these are listed below:

1. Line 1 - Consider removing "Treatment" from the title: the study's primary objective was to report on mortality and not treatment

2. Line 25 - drop "still".

2. Line 33 - drop "by".

3. Line 44 - drop "not giving".

4. Line 45 - drop "no" and combine HiB and PCV vaccine. (This is according to your analysis since you assumed both Hib and PCV were given but this is a significant flaw in your study - refer to comment 11 below)

5. Lines 49 - 50 - Conclusion should be based on your study findings while acknowledging limitaitons. Your study cannot conclude that "ministry of public health should increae coverage". Consider rewording this statement along the lines of "...increasing vaccine coverage can potentially reduce the risk of death from ABM in children in Afghanistan..."

6. Lines 62 - 68: move these sentences ("In 2016.." to "respectively[5]") to the next paragraph.

7. Line 81: Add "The" before "Afghanistan".

8. Line 85: Add "The" before "main".

9. Lines 92-93: Author needs to specify which variables were obtained from interviewing parents/guardians and which ones were obtained by abstraction from clinical records.

10. Lines 96-98: AS mentioned above, indicate which of the above variables were based on physician documentation in clinical records.

11. Lines 102-103: How did authors determine whether patients had received both Hib and PCV vaccines? Even though given at the same age, it is possible that some might have missed one or the other. Also, note according to lines 82-83 PCV-13 was introduced 4 years after Hib, therefore it is not correct to assume that because a child received a vaccine at 6, 10 or 14 weeks they received the Hib and PCV-13. Also, can you please explain how you accounted for those who did not complete vaccination series e.g. if they only received 1 or 2 doses.

12. Line 107: Add "were" before willing.

13. Lines 111-112: Drop these lines; not required in a manuscript

14. Line 114: Drop "treatment"

15. Line 117: Consider dropping "risk"

16. Line 118: Add "children with" before "ABM" and drop "patients".

17. Lines 120-124: Is the WHO criteria used by said pediatricians in their routine practice or were they instructed to apply these during study period?

18. Line 121: consider dropping "expert"

19. Line 125: Drop "Both male and female"

20. Lines 128-131: Need to explain who made this diagnosis. Was it the attending paediatrician? or was this determined by study investigators? Also what is the name of this score? Is this score used in routine practice or was this specifically used during the study period?

21. Lines 132-1314 Need to explain who made this diagnosis. Was it the attending paediatrician? or was this determined by study investigators? Also is the criteria used in routine practice or was this specifically used during the study period?

22. Line 138: It is not clear why macrolides were excluded from this study? Macrolides are unlikely to have an effect on CNS infections.

23. Lines 170-171: Move the sentence on adjunctive dexa therapy to the paragraph of table 4 i.e. Lines 197-201

24. Line 176 (Table 1): Move the row of age(mean) to the row with the age categories. Also present the median(IQR) for age as it is unlikely you have a normal distribution.

25. Line 185 (Table 2): Headache - in the methods section please explain why the N for headaches is less than the total. Explain that it was limited to those whom headache could be assessed.

26. Line 185 (Table 2): No. of concurrent infections - consider dropping this row. This does not add much value to the study and it is extremely unusual to have two or more true infections besides meningitis.

27. Line 186: Coma should be defined using clinical criteria i.e. based on GCS or AVPU scale

28. Line 194 (Table 3): What was the N for this table?

29. Line 197-199: Move the lines of concurrent infections to the write up of Table 1 (from lines 161-172)

30. Line 203 (Table 4): Exclude the self-discharged group entirely from this table. All the variables in this table are dependent on the patient being admitted and therefore cannot be included here. The exclusion of self-discharged group from this part of the analysis should be stated in the methodology.

31. Line 203 (Table 4): Drop the number of complications from this tables; not as important or of any value to the study.

32. Line 207: Drop "main" and add "significantly" before "associated".

33. Line 215 (Table 5): The reference groups indicated here are mostly not correct. E.g. check Age, PCV, Hib, Vomitting, irritability.

34. Line 215 (Table 5): The percentages presented for the survived and died groups for each row should be the proportions based on row totals and not column totals. E.g. Age <5 survived 129 (82.7%) died (17.3%), Age >=5 survived (74.5%) died (25.5%).

35. Line 226 (Table 6): There is no need to present the numbers of patients who survived or died in each row. It might be better to include a column of the Crude OR in this table, as this allows to compare the crude and adjusted at the same instance.

36. Line 238-239: Consider dropping the Canada study, as the age of the study population is not the same

37. Line 240-243: Consider dropping the UK & Ireland, France, and Taiwan study, as the as the age of the study population is not the same.

38. Line 247: Consider dropping the Saudi Arabia study as the age is also not the same

39. Line 252-254: Is there a reference to this statement?

40. Line 270: change from "is declined" to "has declined"

41. Line 282: drop "is" before "may"

7. PLOS authors have the option to publish the peer review history of their article (what does this mean?). If published, this will include your full peer review and any attached files.

Reviewer #1: **Yes: **Adeel A Shah

---

## [Author Response · Author response to Decision Letter 2]

27 Oct 2021

First of all, I am very thankful of the useful comments of the reviewer. I really appreciate it. As we have very less facilities (especially in research) in war-torn Afghanistan, we have tried our best to show the world a real picture of pediatric acute bacterial meningitis in Kandahar, Afghanistan. But I agree that there were many limitations in our study. Thanks again for your kind and fruitful comments.

6. Review Comments to the Author

Reviewer #1: Overall ,the authors have done a great job in improving their manuscript. They must be commended for taking their time in addressing each and every single issue that had been raised and have done so with great detail.

Nevertheless, there are still some issues that need to be addressed and these are listed below:

1. Line 1 - Consider removing "Treatment" from the title: the study's primary objective was to report on mortality and not treatment.

Answer: OK. Removed.

2. Line 25 - drop "still".

Answer: OK. Dropped.

2. Line 33 - drop "by".

Answer: OK. Dropped.

3. Line 44 - drop "not giving".

Answer: OK. Dropped.

4. Line 45 - drop "no" and combine HiB and PCV vaccine. (This is according to your analysis since you assumed both Hib and PCV were given but this is a significant flaw in your study - refer to comment 11 below)

Answer: OK. Done.

5. Lines 49 - 50 - Conclusion should be based on your study findings while acknowledging limitaitons. Your study cannot conclude that "ministry of public health should increae coverage". Consider rewording this statement along the lines of "...increasing vaccine coverage can potentially reduce the risk of death from ABM in children in Afghanistan..."

Answer: OK. Done. Conclusion sentence has been changed.

6. Lines 62 - 68: move these sentences ("In 2016.." to "respectively[5]") to the next paragraph.

Answer: OK. Done.

7. Line 81: Add "The" before "Afghanistan".

Answer: OK. Done.

8. Line 85: Add "The" before "main".

Answer: OK. Done.

9. Lines 92-93: Author needs to specify which variables were obtained from interviewing parents/guardians and which ones were obtained by abstraction from clinical records.

Answer: OK. Done as follows: Data of the patients admitted with the diagnosis of ABM werewas collected during 12-month-period (February 2020–January 2021) using questionnaire with questions variables regarding socio-demographic characteristics, and physical signs and symptoms obtained from physical examination or interviewing the parent/guardian of the patients; while , laboratory examinations, and treatment variables extracted from clinical records of the patients.

10. Lines 96-98: AS mentioned above, indicate which of the above variables were based on physician documentation in clinical records.

Answer: OK. Done.

11. Lines 102-103: How did authors determine whether patients had received both Hib and PCV vaccines? Even though given at the same age, it is possible that some might have missed one or the other. Also, note according to lines 82-83 PCV-13 was introduced 4 years after Hib, therefore it is not correct to assume that because a child received a vaccine at 6, 10 or 14 weeks they received the Hib and PCV-13. Also, can you please explain how you accounted for those who did not complete vaccination series e.g. if they only received 1 or 2 doses.

Answer: Yes, we totally agree with your kind comment. Our main confirmation of the Hib and PCV vaccines were based on seeing the EPI card of the child. If card was not available, our decision was based on the recall of the parents or guardians, which we agree that can be biased. 

We tried our best to be accurate as possible, but we admit that we could not avoid some of the main problems, like re-call bias. 

12. Line 107: Add "were" before willing.

Answer: OK. Added.

13. Lines 111-112: Drop these lines; not required in a manuscript

Answer: OK. Dropped.

14. Line 114: Drop "treatment"

Answer: OK. Dropped.

15. Line 117: Consider dropping "risk"

Answer: OK. Dropped.

16. Line 118: Add "children with" before "ABM" and drop "patients".

Answer: OK. Added.

17. Lines 120-124: Is the WHO criteria used by said pediatricians in their routine practice or were they instructed to apply these during study period?

Answer: Yes, the pediatricians were routinely using the WHO criteria.

18. Line 121: consider dropping "expert"

Answer: OK. Dropped.

19. Line 125: Drop "Both male and female"

Answer: OK. Dropped.

20. Lines 128-131: Need to explain who made this diagnosis. Was it the attending paediatrician? or was this determined by study investigators? Also what is the name of this score? Is this score used in routine practice or was this specifically used during the study period?

Answer: OK. Explained. This score of diagnosis is routinely used by pediatricians in the hospital. First on admission diagnosis is made by the pediatricians, then our investigators were re-doing it for the inclusion in our study.

21. Lines 132-1314 Need to explain who made this diagnosis. Was it the attending paediatrician? or was this determined by study investigators? Also is the criteria used in routine practice or was this specifically used during the study period?

Answer: OK. Explained. This score of diagnosis is routinely used by pediatricians in the hospital. First on admission diagnosis is made by the pediatricians, then our investigators were re-doing it for the inclusion in our study.

22. Line 138: It is not clear why macrolides were excluded from this study? Macrolides are unlikely to have an effect on CNS infections.

Answer: Yes, we agree. It is corrected to “Penicillins”. 

23. Lines 170-171: Move the sentence on adjunctive dexa therapy to the paragraph of table 4 i.e. Lines 197-201

Answer: OK. Done .

24. Line 176 (Table 1): Move the row of age(mean) to the row with the age categories. Also present the median(IQR) for age as it is unlikely you have a normal distribution.

Answer: OK. Done.

25. Line 185 (Table 2): Headache - in the methods section please explain why the N for headaches is less than the total. Explain that it was limited to those whom headache could be assessed.

Answer: OK. Done. 

26. Line 185 (Table 2): No. of concurrent infections - consider dropping this row. This does not add much value to the study and it is extremely unusual to have two or more true infections besides meningitis.

Answer: Ok. Done.

27. Line 186: Coma should be defined using clinical criteria i.e. based on GCS or AVPU scale

Answer: OK. Done.

28. Line 194 (Table 3): What was the N for this table?

Answer: It is 393. Now added.

29. Line 197-199: Move the lines of concurrent infections to the write up of Table 1 (from lines 161-172)

Answer: OK. Done. It belongs to write up of table 2, so I moved it to Table 2.

30. Line 203 (Table 4): Exclude the self-discharged group entirely from this table. All the variables in this table are dependent on the patient being admitted and therefore cannot be included here. The exclusion of self-discharged group from this part of the analysis should be stated in the methodology.

Answer: OK. Done. All the table is re-made based on your comments.

31. Line 203 (Table 4): Drop the number of complications from this tables; not as important or of any value to the study.

Answer: OK. Done. 

32. Line 207: Drop "main" and add "significantly" before "associated".

Answer: OK. Done.

33. Line 215 (Table 5): The reference groups indicated here are mostly not correct. E.g. check Age, PCV, Hib, Vomitting, irritability.

Answer: OK. Done. All reference values have been corrected.

34. Line 215 (Table 5): The percentages presented for the survived and died groups for each row should be the proportions based on row totals and not column totals. E.g. Age <5 survived 129 (82.7%) died (17.3%), Age >=5 survived (74.5%) died (25.5%).

Answer: OK. Done. All the percentages have been changed based on your comments.

35. Line 226 (Table 6): There is no need to present the numbers of patients who survived or died in each row. It might be better to include a column of the Crude OR in this table, as this allows to compare the crude and adjusted at the same instance.

Answer: OK. Done. Numbers of survived and dead have been removed. Also, a new column of Crude OR has been added.

36. Line 238-239: Consider dropping the Canada study, as the age of the study population is not the same

Answer: OK. Removed.

37. Line 240-243: Consider dropping the UK & Ireland, France, and Taiwan study, as the as the age of the study population is not the same.

Answer: OK. Removed.

38. Line 247: Consider dropping the Saudi Arabia study as the age is also not the same

Answer: OK. Removed.

39. Line 252-254: Is there a reference to this statement?

Answer: Sorry, unfortunately “No”. This information is based on the experience of our research team and pediatricians of the hospital. This is a very common problem.

40. Line 270: change from "is declined" to "has declined"

Answer: Ok. Done.

41. Line 282: drop "is" before "may"

Answer: OK. Done.

Many thanks for your kind comments.

---

## [Decision Letter · Decision Letter 3]

6 Dec 2021

PONE-D-21-20289R3Treatment outcome of acute bacterial meningitis among children in Kandahar, Afghanistan: A prospective study.PLOS ONE

Dear Dr. Rahimi,

Thank you for submitting your manuscript to PLOS ONE. After careful consideration, we feel that it has merit but does not fully meet PLOS ONE’s publication criteria as it currently stands. Therefore, we invite you to submit a revised version of the manuscript that addresses the points raised during the review process.

The major issues now are related to grammar in addition to a few other items that need attention. I would strongly urge you to have a native English speaker review the manuscript English usage in order to facilitate the publication of this manuscript. Please submit your revised manuscript by Jan 20 2022 11:59PM. If you will need more time than this to complete your revisions, please reply to this message or contact the journal office at plosone@plos.org. Please include the following items when submitting your revised manuscript:A rebuttal letter that responds to each point raised by the academic editor and reviewer(s). You should upload this letter as a separate file labeled 'Response to Reviewers'.A marked-up copy of your manuscript that highlights changes made to the original version. You should upload this as a separate file labeled 'Revised Manuscript with Track Changes'.An unmarked version of your revised paper without tracked changes. You should upload this as a separate file labeled 'Manuscript'.If applicable, we recommend that you deposit your laboratory protocols in protocols.io to enhance the reproducibility of your results. Protocols.io assigns your protocol its own identifier (DOI) so that it can be cited independently in the future. For instructions see: https://journals.plos.org/plosone/s/submission-guidelines#loc-laboratory-protocols. Additionally, PLOS ONE offers an option for publishing peer-reviewed Lab Protocol articles, which describe protocols hosted on protocols.io. Read more information on sharing protocols at https://plos.org/protocols?utm_medium=editorial-email&utm_source=authorletters&utm_campaign=protocols.

We look forward to receiving your revised manuscript.

Kind regards,

Rodney D Adam

Academic Editor

PLOS ONE

Journal Requirements:

Reviewers' comments:

Reviewer's Responses to Questions

**Comments to the Author**

1. If the authors have adequately addressed your comments raised in a previous round of review and you feel that this manuscript is now acceptable for publication, you may indicate that here to bypass the “Comments to the Author” section, enter your conflict of interest statement in the “Confidential to Editor” section, and submit your "Accept" recommendation.

Reviewer #1: (No Response)

2. Is the manuscript technically sound, and do the data support the conclusions?

Reviewer #1: Partly

3. Has the statistical analysis been performed appropriately and rigorously? 

Reviewer #1: Yes

4. Have the authors made all data underlying the findings in their manuscript fully available?

Reviewer #1: Yes

5. Is the manuscript presented in an intelligible fashion and written in standard English?

Reviewer #1: Yes

6. Review Comments to the Author

Reviewer #1: The authors have done a good job in addressing the previous comments and the quality of the manuscript is greatly improved.

There are some minor issues that have been noted at this time and these need to be addressed so as to improve the readability and coherence of the manuscript:

1. Lines 39-42: rewrite this sentence - "Concurrent infections..."; has logical and grammatical errors

2. Lines 47-48: consider removing the sentence "The coverage of PCV and Hib...", can be incorporated with the sentence in line 49 - "Increasing vaccine coverage..."

3. Line 56: change from "the case fatality" to "a case fatality"

4. Line 57: rewrite to "7% and a risk of neurological..."

5. Line 72: use number formatting for the figures presented i.e. 63,001; 16,335; 7,302 and 6,907

6. Lines 83-84: rewrite the sentence from "Unfortunately, there is currently..." to "Currently, there is no published data on the outcome of ABM in children in Afghanistan and the factors associated with poor outcomes."

7. Line 85: remove the word "treatment" and also reword "outcome of ABM patients.." to "outcome of ABM in children..."

8. Line 91: should read "during a 12-month period"

9. Line 109: Change to Study design and participants, then move the study design and period to be part of this section

10. Lines 92-108: this should be under a section called "Study procedures" and should come after the Study design and participants section

11. Lines 147-148: remove this section altogether; no added value of having this presented in the manuscript.

12. Line 173: Median of 4.8 seems incorrect i.e. median is usually a whole number or half of two whole numbers e.g. 4.5 if the median lies between 4 and 5; please check again how you calculated this, as 4.8 would be more of mean than median.

12. Line 173: 207 (52.7%) children under age 5, this is discrepant with the data presented in table 1 - please check

13. Line 182: Table 1:

- median (IQR) - present the IQR as a range i.e. the 25th and 75th centile figures rather than the

actual value of the range e.g. for age it would be median (IQR) - 4.5(2.5-7); (please do this

wherever IQR is presented i.e. Table 3)

- consider combining HiB and PCV as one category - since you mentioned in your previous response

that you confirmed everyone who received HiB also received PCV. (also do this for all the other

tables where you present Hib and PCV data i.e. Tables 5 and 6)

14. Lines 184 - 185: some of the footnotes not relevant to this table e.g. cm, CSF, mg/dl, kg

15. Line 200: you refer to mean +/- SD while in the table it is median (IQR); please confirm which it is and correct accordingly

16. Line 236: Table 6 - please re-look at the adjusted OR for purple rash, this seems to be an error which is most likely due to coding; this kind of dramatic reversal cannot be explained by the other variables in the model.

17. Line 244: drop "living in over-crowded houses, presence of complications" as this was not assessed in the univariate analysis.

18. Line 247: "change from in-hospital mortality rate" to "in-hospital case fatality" and also correct spelling of "Compared"

19. Line 255: consider dropping "huge"

20. Lines 259-261: consider dropping this entire sentence: "The in-hospital mortality...better treatment". This is because you have dealt with the self-discharge group in the outcome analysis and the 21.5% fatality was of the group that did not include the self-discharged group.

21. Line 305: change from "mortality rate" to "case fatality"

22. Line 307: drop "living in over-crowded houses, presence of complications" as this was not assessed in the univariate analysis.

23. Line 313: UNICEF should be in all caps.

7. PLOS authors have the option to publish the peer review history of their article (what does this mean?). If published, this will include your full peer review and any attached files.

Reviewer #1: **Yes: **Adeel Ahmad Shah

---

## [Author Response · Author response to Decision Letter 3]

11 Jan 2022

First of all, I am very thankful of the useful comments of the reviewer.

As most of the comments were related to English language grammar, so I sent the manuscript to a native speaker of English. Many thanks to him, as he corrected the whole manuscript grammatically.

Just one thing to remind you that due to thorough grammatical review, the line numbers have changed from the preview version.

Thanks again for your kind and fruitful comments.

Following are the comments, followed by my answers:

6. Review Comments to the Author

Reviewer #1: The authors have done a good job in addressing the previous comments and the quality of the manuscript is greatly improved.

There are some minor issues that have been noted at this time and these need to be addressed so as to improve the readability and coherence of the manuscript:

1. Lines 39-42: rewrite this sentence - "Concurrent infections..."; has logical and grammatical errors.

Answer: OK. Now this sentence (as well as the whole manuscript) is reviewed and corrected by a native English speaker. 

2. Lines 47-48: consider removing the sentence "The coverage of PCV and Hib...", can be incorporated with the sentence in line 49 - "Increasing vaccine coverage..."

Answer: OK done. Now the sentence "The coverage of PCV and Hib..." is dropped.

3. Line 56: change from "the case fatality" to "a case fatality"

Answer: OK done. It is changed accordingly.

4. Line 57: rewrite to "7% and a risk of neurological..."

Answer: Ok done. Now it is re-written and corrected by a native English speaker.

5. Line 72: use number formatting for the figures presented i.e. 63,001; 16,335; 7,302 and 6,907

Answer: OK done. Now it is corrected as per your advice. 

6. Lines 83-84: rewrite the sentence from "Unfortunately, there is currently..." to "Currently, there is no published data on the outcome of ABM in children in Afghanistan and the factors associated with poor outcomes."

Answer: OK done. Now it is re-written and corrected by a native English speaker.

7. Line 85: remove the word "treatment" and also reword "outcome of ABM patients.." to "outcome of ABM in children..."

Answer: OK done. Now corrected as per your advice.

8. Line 91: should read "during a 12-month period"

Answer: OK done.

9. Line 109: Change to Study design and participants, then move the study design and period to be part of this section

Answer: OK done.

10. Lines 92-108: this should be under a section called "Study procedures" and should come after the Study design and participants section

Answer: OK done.

11. Lines 147-148: remove this section altogether; no added value of having this presented in the manuscript.

Answer: OK done.

12. Line 173: Median of 4.8 seems incorrect i.e. median is usually a whole number or half of two whole numbers e.g. 4.5 if the median lies between 4 and 5; please check again how you calculated this, as 4.8 would be more of mean than median.

Answer: Sorry. Thanks for the very good point. Yes, you are right. It is mean (SD). Now I corrected it in the text and table.

12. Line 173: 207 (52.7%) children under age 5, this is discrepant with the data presented in table 1 - please check

Answer: Sorry. Thanks for the very good point. Yes, you are right. Now I corrected it.

13. Line 182: Table 1:

- median (IQR) - present the IQR as a range i.e. the 25th and 75th centile figures rather than the

actual value of the range e.g. for age it would be median (IQR) - 4.5(2.5-7); (please do this

wherever IQR is presented i.e. Table 3)

- consider combining HiB and PCV as one category - since you mentioned in your previous response

that you confirmed everyone who received HiB also received PCV. (also do this for all the other

tables where you present Hib and PCV data i.e. Tables 5 and 6)

Answer: OK thanks. Done. IQR are now presented as a range i.e. the 25th and 75th centile figures. Also, PCV and Hib have been combined as one category.

14. Lines 184 - 185: some of the footnotes not relevant to this table e.g. cm, CSF, mg/dl, kg

Answer: Many thanks for the good point. Now I have removed the irrelevant ones.

15. Line 200: you refer to mean +/- SD while in the table it is median (IQR); please confirm which it is and correct accordingly

Answer: Thanks. Sorry for that. It was median (IQR). Now corrected.

16. Line 236: Table 6 - please re-look at the adjusted OR for purple rash, this seems to be an error which is most likely due to coding; this kind of dramatic reversal cannot be explained by the other variables in the model.

Answer: Thanks for that. Data was re-analyzed. There was in fact an error in coding. Now new corrected values have been written, i.e., COR (95% CI) is 4.0 (1.5-10.6), AOR (95%CI) is 3.1 (0.9-8.3), and the p-value is 0.05.

17. Line 244: drop "living in over-crowded houses, presence of complications" as this was not assessed in the univariate analysis.

Answer: OK. Now "living in over-crowded houses, presence of complications" is dropped.

18. Line 247: "change from in-hospital mortality rate" to "in-hospital case fatality" and also correct spelling of "Compared"

Answer: OK. Now correct as per your advice.

19. Line 255: consider dropping "huge"

Answer: OK. Now “huge” is removed.

20. Lines 259-261: consider dropping this entire sentence: "The in-hospital mortality...better treatment". This is because you have dealt with the self-discharge group in the outcome analysis and the 21.5% fatality was of the group that did not include the self-discharged group.

Answer: OK. Now this sentence is dropped.

21. Line 305: change from "mortality rate" to "case fatality"

Answer: OK. Now changed.

22. Line 307: drop "living in over-crowded houses, presence of complications" as this was not assessed in the univariate analysis.

Answer: OK. Now "living in over-crowded houses, presence of complications" is dropped.

23. Line 313: UNICEF should be in all caps.

Answer: Ok. Now done.

Many thanks for your kind comments.

---

## [Editor Report · Decision Letter 4]

2 Feb 2022

PONE-D-21-20289R4Outcome of acute bacterial meningitis among children in Kandahar, Afghanistan: A prospective observational cohort study.PLOS ONE

Dear Dr. Rahimi,

Thank you for submitting your manuscript to PLOS ONE. After careful consideration, we feel that it has merit but does not fully meet PLOS ONE’s publication criteria as it currently stands. Therefore, we invite you to submit a revised version of the manuscript that addresses the points raised during the review process.

I have gone through the reviewer comments and responses and have a number of issues that must still be addressed. The authors must remember that there is no copy editor so the manuscript must be ready to go when approved.

The major substantive issue remaining is the CSF white count. What lab method was used? If it was the hemocytometer, mononuclear cells are lumped together (rather than a specific reference to lymphocytes). Thus, the percent of neutrophils and mononuclears should add up to 100%. That does not appear to be the case for the data shown in Table 3. It is also highly unusual for acute bacterial meningitis to have a mononuclear predominance as the data in Table 3 suggest; uncommon enough to require re-evaluation of the presumptive diagnosis of bacterial meningitis.

The follow up is not available for the 72 who “discharged themselves” from the hospital. I assume that means that the parents removed them against medical advice. Is anything known about their outcome? Especially, how many were removed because the parents thought they were dying?

Line 30: “of” after total is missing. Also, median has been used instead of mean

Line 58: a citation is needed at the end of the first sentence

Line 156: the ^0^C did not come through correctly on my version of the pdf

Line 158: eliminate the space before the comma

We look forward to receiving your revised manuscript.

Kind regards,

Rodney D Adam

Academic Editor

PLOS ONE
---

## [Author Response · Author response to Decision Letter 4]

8 Feb 2022

First of all, I am very thankful of the useful comments of the reviewer.

Following are the comments, followed by my answers:

Comment 1: The major substantive issue remaining is the CSF white count. What lab method was used? If it was the hemocytometer, mononuclear cells are lumped together (rather than a specific reference to lymphocytes). Thus, the percent of neutrophils and mononuclears should add up to 100%. That does not appear to be the case for the data shown in Table 3. It is also highly unusual for acute bacterial meningitis to have a mononuclear predominance as the data in Table 3 suggest; uncommon enough to require re-evaluation of the presumptive diagnosis of bacterial meningitis.

Answer: First of all, thanks a lot for the very good and important comment. 

Yes, lab used hemocytometer method for CSF white cells. You are right, they were mononuclear cells (not lymphocytes). Now lymphocytes have been changed to “mononuclear cells”. 

To answer second part of your comment, we re-analyzed all the data. Unfortunately, there were mistakes in the data analysis of CSF white counts. We are very sorry for this. The data in Table 3 and text were corrected based on the re-analysis.

Once again, many thanks for the extremely useful comment. Also, accept our apologies for the mistakes. 

Comment 2: The follow up is not available for the 72 who “discharged themselves” from the hospital. I assume that means that the parents removed them against medical advice. Is anything known about their outcome? Especially, how many were removed because the parents thought they were dying?

Answer: These 72 children were discharged by their parents against medical advice. As their parents did not ask us before discharge (also, they did not tell us the reason for the self-discharge), so we could not follow them up and did not know about their outcome. In conclusion, we do not know the exact reason of their self-discharge. We know just one thing that most of these 72 LAMA patients were in critical situation and possibly their parents thought they were dying and took their children to Kabul or another country (especially Pakistan or India) for better treatment.

Line 30: “of” after total is missing. Also, median has been used instead of mean

Answer: OK. Done. “of” is added while median is changed to mean.

Line 58: a citation is needed at the end of the first sentence

Answer: OK. Citation is added.

Line 156: the 0C did not come through correctly on my version of the pdf

Answer: OK. Done. Degree centigrade symbol is corrected.

Line 158: eliminate the space before the comma

Answer: OK. Done. Space is removed.

As per your comment, "Funding" information has been removed from the manuscript.

Please put the following information in the online submission of my manuscript:

Funding: No specific funding was obtained for this study, but WR Taylor is part funded by Wellcome under grant 220211. For the purpose of Open Access, the author has applied a CC BY public copyright licence to any Author Accepted Manuscript version arising from this submission.

Many thanks for your kind and useful comments.

---

## [Editor Report · Decision Letter 5]

28 Feb 2022

PONE-D-21-20289R5Outcome of acute bacterial meningitis among children in Kandahar, Afghanistan: A prospective observational cohort study.PLOS ONE

Dear Dr. Rahimi,

Thank you for submitting your manuscript to PLOS ONE. After careful consideration, we feel that it has merit but does not fully meet PLOS ONE’s publication criteria as it currently stands. Therefore, we invite you to submit a revised version of the manuscript that addresses the points raised during the review process.

 Thank you for your revision. There are still two issues that should be covered before publication:1. The fact that 72 of the children were critical ill when brought home by their parents means that the mortality rate may have been much higher than the stated 21.5%. If all of these died, the mortality would have actually been 35.9%. It would be appropriate to raise this possibility at the beginning of the discussion where the mortality rate is discussed.2. I am surprised by the high percent of CSFs with a mononuclear predominance (25%). Is it possible that some of these were actually tuberculous meningitis? That possibility should also be raised in the discussion.

We look forward to receiving your revised manuscript.

Kind regards,

Rodney D Adam

Academic Editor

PLOS ONE
---

## [Author Response · Author response to Decision Letter 5]

2 Mar 2022

First of all, I am very thankful of the useful comments of the reviewer.

Following are the comments, followed by my answers:

Thank you for your revision. There are still two issues that should be covered before publication:

1. The fact that 72 of the children were critical ill when brought home by their parents means that the mortality rate may have been much higher than the stated 21.5%. If all of these died, the mortality would have actually been 35.9%. It would be appropriate to raise this possibility at the beginning of the discussion where the mortality rate is discussed.

Answer: OK. Done. This important comment is added to the discussion.

2. I am surprised by the high percent of CSFs with a mononuclear predominance (25%). Is it possible that some of these were actually tuberculous meningitis? That possibility should also be raised in the discussion.

Answer: OK. Done. The possibility of some TBM patients has been raised in the discussion part of the manuscript.

Many thanks for your kind and useful comments.

---

## [Editor Report · Decision Letter 6]

3 Mar 2022

Outcome of acute bacterial meningitis among children in Kandahar, Afghanistan: A prospective observational cohort study.

PONE-D-21-20289R6

Dear Dr. Rahimi,

We’re pleased to inform you that your manuscript has been judged scientifically suitable for publication and will be formally accepted for publication once it meets all outstanding technical requirements.

Kind regards,

Rodney D Adam

Academic Editor

PLOS ONE
---

## [Editor Report · Acceptance letter]

31 Mar 2022

PONE-D-21-20289R6 

Outcome of acute bacterial meningitis among children in Kandahar, Afghanistan: A prospective observational cohort study. 

Dear Dr. Rahimi:

I'm pleased to inform you that your manuscript has been deemed suitable for publication in PLOS ONE. Congratulations! Your manuscript is now with our production department. 

Kind regards, 

on behalf of

Dr. Rodney D Adam 

Academic Editor

PLOS ONE